# Foxo3 Knockdown Mediates Decline of *Myod1* and *Myog* Reducing Myoblast Conversion to Myotubes

**DOI:** 10.3390/cells12172167

**Published:** 2023-08-29

**Authors:** Benjamin Gellhaus, Kai O. Böker, Marlene Gsaenger, Eyck Rodenwaldt, Marc A. Hüser, Arndt F. Schilling, Dominik Saul

**Affiliations:** 1Department of Trauma Surgery, Orthopedics and Plastic Surgery, University Medical Center Goettingen, Robert-Koch-Str. 40, 37099 Göttingen, Germany; benjamin.gellhaus@stud.uni-goettingen.de (B.G.); kai.boeker@med.uni-goettingen.de (K.O.B.); eyck.rodenwaldt@med.uni-goettingen.de (E.R.); arndt.schilling@med.uni-goettingen.de (A.F.S.); 2Department of Otorhinolaryngology, Head and Neck Surgery, University Medical Center Goettingen, Robert-Koch-Str. 40, 37099 Göttingen, Germany; marc.hueser@med.uni-goettingen.de; 3Department of Trauma and Reconstructive Surgery, Eberhard Karls University Tübingen, BG Trauma Center Tübingen, 72076 Tübingen, Germany; 4Kogod Center on Aging and Division of Endocrinology, Mayo Clinic, Rochester, MN 55905, USA

**Keywords:** sarcopenia, skeletal muscle, Foxo3, Tp53, Myod1, Myog

## Abstract

Sarcopenia has a high prevalence among the aging population. Sarcopenia is of tremendous socioeconomic importance because it can lead to falls and hospitalization, subsequently increasing healthcare costs while limiting quality of life. In sarcopenic muscle fibers, the E3 ubiquitin ligase F-Box Protein 32 (Fbxo32) is expressed at substantially higher levels, driving ubiquitin-proteasomal muscle protein degradation. As one of the key regulators of muscular equilibrium, the transcription factor Forkhead Box O3 (FOXO3) can increase the expression of Fbxo32, making it a possible target for the regulation of this detrimental pathway. To test this hypothesis, murine C2C12 myoblasts were transduced with AAVs carrying a plasmid for four specific siRNAs against Foxo3. Successfully transduced myoblasts were selected via FACS cell sorting to establish single clone cell lines. Sorted myoblasts were further differentiated into myotubes and stained for myosin heavy chain (MHC) by immunofluorescence. The resulting area was calculated. Myotube contractions were induced by electrical stimulation and quantified. We found an increased Foxo3 expression in satellite cells in human skeletal muscle and an age-related increase in Foxo3 expression in older mice in silico. We established an in vitro AAV-mediated FOXO3 knockdown on protein level. Surprisingly, the myotubes with FOXO3 knockdown displayed a smaller myotube size and a lower number of nuclei per myotube compared to the control myotubes (AAV-transduced with a functionless control plasmid). During differentiation, a lower level of FOXO3 reduced the expression Fbxo32 within the first three days. Moreover, the expression of Myod1 and Myog via ATM and Tp53 was reduced. Functionally, the Foxo3 knockdown myotubes showed a higher contraction duration and time to peak. Early Foxo3 knockdown seems to terminate the initiation of differentiation due to lack of Myod1 expression, and mediates the inhibition of Myog. Subsequently, the myotube size is reduced and the excitability to electrical stimulation is altered.

## 1. Introduction

Low muscle strength is the major criterion used to define sarcopenia according to the European Working Group on Sarcopenia in Older People (EWGSOP2). If accompanied by a lower muscle quality and quantity, the diagnosis “sarcopenia” is confirmed. When combined with a low physical performance, severe sarcopenia is assumed [1]. With an estimated prevalence of 35% in patients admitted to the hospital, another 15% of non-sarcopenic patients develop sarcopenia until discharge [2], highlighting not just the importance of sarcopenia in the general population, but particularly in hospitalized patients.

In general, sarcopenia is associated with an increased rate of falls [3], prolonged hospitalization [4] and a limited quality of life [5]. Economically, sarcopenia caused hospitalization costs of $40.4 billion in the US in 2014, making sarcopenia both an economically and socially impactful disease [6].

The clinical relevance raises the question of current therapies and their impact. Physical activity and nutritional supplementation have been suggested. Moderate intensity physical activity has been shown to increase the ability to walk 400 m in older people (70–89 years) over a training period of 2.6 years [7]. Similarly, 12 months of high intensity weight lifting training significantly reduced the mortality and nursing home admission in hip fracture patients [8]. Physical activity is effective, but requires long time perseverance and may be difficult to achieve in hospitalized patients because the majority of hospital time (>80%) is spent in bed [9]. Nutritional supplementation, such as vitamin D and leucine-enriched whey protein, also requires prolonged use (13 weeks), and has shown beneficial effects only on appendicular muscle mass, while failing to improve handgrip strength in older adults ( ≥65 years) [10]. Novel molecular therapeutics that specifically address the molecular cause of sarcopenia may be able to improve muscle function over shorter periods of time.

In sarcopenic muscle, at the molecular level, anabolic conditions lead to high levels of insulin-like growth factor 1 (IGF-1), which causes receptor tyrosine kinases to recruit phosphoinositide 3-kinases (PI3K) to convert phosphatidylinositol-4,5-bisphosphate (PIP2) to phosphatidylinositol-3,4,5-trisphosphate (PIP3). The serine–threonine kinase AKT and phosphoinositide-dependent kinase 1 (PDK1) are then recruited, resulting in AKT activation. AKT phosphorylates the Forkhead Box O3 (FOXO3), thereby preventing its entry into the nucleus and inhibiting its function as a transcription factor [11]. FOXO3 is a member of the forkhead transcription factor family, which share a characteristic forkhead DNA-binding domain. In particular, the FoxO subfamily mediates the insulin signaling and consequently glucose hemostasis [12]. Briefly, IGF-1 suppresses FOXO3 and subsequently its target gene *Fbxo32* (also known as *Atrogin-1*) via upregulation of the PI3K/AKT pathway [13,14,15,16]. *Fbxo32* was previously observed to be selectively expressed in cardiac fibers and skeletal muscle, with higher expression in atrophic muscle [17].

We hypothesize that a reduction of *Fbxo32* improves muscle fiber size and contractility and has the potential to selectively target skeletal muscle fibers to increase muscle strength. In the current study, we sought to determine the morphological and functional consequences of a *Foxo3* knockdown in differentiating C2C12 myoblasts. We hypothesize that a *Foxo3* knockdown leads to a reduction of its target gene *Fbxo32,* resulting in an anabolic effect on muscle cells, larger cell bodies and improved contractility.

## 2. Material and Methods

### 2.1. Cell Culture

C2C12 (CRL-1772) cells were purchased from American Type Culture Collection (ATCC, Manassas, VA, USA) and cultured at 37 °C in 5% CO_2_ and 95% humidity. To induce the differentiation into myotubes, growth medium Dulbecco’s Modified Eagle Medium (DMEM, Thermo Fisher Scientific, MA, USA) including 10% FCS (Pan-Biotech Gmbh, Aidenbach, Germany) and 1% Penicillin/Streptomycin (P/S, Pan Biotech, Aidenach, Germany) was replaced by low serum differentiation medium (DMEM including 2% Horse Serum (HS; Sigma-Aldrich, St. Louis, MO, USA) and 1% P/S). A total of 60.000 C2C12 myoblasts were seeded on a 24-well plate incubated 48h hours prior to differentiation. The time of initial differentiation is defined as day 1. C2C12 myoblasts were either transfected with Lipofectamine RNAiMAX (Thermo Fisher, MA, USA) carrying *Foxo3* siRNA (100380, Ambion^®^ by Life Technologies, Carlsbad, CA, USA), transduced with AAV-siRNA/GFP against *Foxo3* mRNA (207900940210, Applied Biological Materials Inc., Richmond, BC, Canada) or functionless siRNA (iAAV01500, Applied Biological Materials Inc.) as control.

### 2.2. Cell Line Selection

Transduced C2C12 myoblasts were selected by BD FACSAria (BD Bioscience, NJ, USA) to create high knockdown single clones. For this, transduced C2C12 myoblasts were harvested by trypsinization, washed with PBS, centrifuged at 300× *g* for 5 min and dissolved in 500 µL growth medium. GFP-positive myoblasts were selected by FACS cell sorting into a 96 well containing 100 µL growth medium. The remaining GFP-positive myoblasts were pooled. Cell lines were identified under fluorescence microscopy and passage step by step from 96 well to T75 flask. Finally, monoclonal *Foxo3* knockdown groups (siFoxo3) and monoclonal (siControl) or polyclonal (siControl pooled) control groups were established. *Foxo3* knockdown efficiency was characterized on a transcriptional level normalized to the level of C2C12 wildtype cells.

### 2.3. Comparison of Growth Speed in-between Selected Cell Lines

For growth speed, cells were cultivated and harvested by trypsinization to quantify the cell number using Countess 3 FL (*n* = 4, Invitrogen, Thermo Fisher Scientific) at different time points. Subsequently, different monoclonal groups of *Foxo3* knockdown (K1, K2, K4 and K6) and control (K8, K9, K10, K11 and K12) were pooled together to represent superior group differences. The growth behavior was approximated by the Gompertz growth model for group differences [18]. For individual consideration, cells were seeded on E-Plate 16 PET (Agilent Technologies, Santa Clara, CA, USA) using the xCELLigence RTCA DP system (Agilent Technologies) to detect the change in electrical resistance as cell index (*n* = 4). Analysis of the area under the curve (*n* = 4) revealed suitable cell lines that were used for further experiments.

### 2.4. Quantitative Real Time PCR

RNA was isolated from C2C12 cells using Trizol^®^ Reagent (Thermo Fisher Scientific). An amount of 1000 ng of RNA were reverse transcribed into cDNA using iScript cDNA Synthesis Kit (Bio-Rad Laboratories, CA, USA) in 20 µL reaction volume. For qPCR, 10 µL SsoAdvanced Universal SYBR Green Supermix were mixed with 0.5 µL of each forward and reverse primer (10 µM) (for sequences see Table 1), and 9 µL of nuclease free H_2_O was applied as a master mix for every sample. Every sample was amplified as a triplicate of each 1 µL cDNA to 19 µL master mix in a 96 well plate in CFX96™ Real-Time Thermal Cycler (Bio-Rad Laboratories). After an initial denaturation at 95 °C for 3 min, 40 cycles of 95 °C for 10 s, 55 °C for 10 s and 72 °C for 30 s were followed by 95 °C for 10 s and a melt curve analysis from 65–95 °C by constantly increasing temperature. The gene expression was calculated using the 2^−∆∆Ct^ method [19]. All samples were measured at least in biological triplicates.

### 2.5. Western Blot

Cells were lysed on ice using RIPA buffer (Tris 250 mM, NaCl 150 mM, SDS 0.1%, Sodium deoxycholate 0.5%, Triton X-100 1% and protease inhibitor (Roche, Switzerland); Merck, Germany, if not labeled differently). The protein sample concentration was detected by BCA assay using Pierce™ BCA Protein Assay Kit (Thermo Fisher Scientific). Sample separation was performed in freshly cast 12% polyacrylamide gels (TGX StainFree™ FastCast™ Acrylamide Kit 12%, Bio-Rad). Protein samples (19 µg) added to an adequate amount of Laemmli loading buffer were separated in SDS-Page and transferred on PVDF using the TransBlot^®^ Turbo™ System (Bio-Rad). PVDF membranes were blocked with 5% milk in TBS-T (Medicago AB, Uppsala, Sweden) at room temperature (RT) for 1 h. Primary antibodies (FOXO3, D19A7, Cell Signaling Technology, MA, USA and ACTB, sc-47778, Santa Cruz Biotechnology, TX, USA) were incubated 1:1000 in 5% milk in TBS-T at 4 °C for 24 h. Secondary antibodies were incubated at RT for 1 h. Images were taken using Clarity™ Western ECL Substrate (Bio-Rad) using ChemiDoc™ XRS+ Imaging System (Bio-Rad) in signal accumulation mode. Band detection and analysis was performed using the Image Lab 6.1.0 software (Bio-Rad). The adjusted band volume was detected as intensity subtracted by the background intensity. The FOXO3 protein level was depicted as relative to the control samples (*n* = 6).

### 2.6. Immunofluorescence Staining

The cells were fixed in 4% PFA (Th. Geyer Gmbh & Co. KG, Renningen, Germany) and washed three times with PBS (Pan-Biotech Gmbh). Samples were blocked and permeabilized in PBS supplemented with 10% HS and 0.2% Triton X-100 (AppliChem GmbH, Darmstadt, Germany). The primary anti-MHC-antibody (1:100) was incubated in PBS +1% HS +0.2% Triton X-100 at 4 °C overnight. On the following day, the secondary antibody (1:100) was incubated in PBS +1% HS for 1 h at RT in the absence of light. Samples were mounted on slides using Fluoroshield™ with DAPI (Sigma-Aldrich). After drying, images were taken using LAS X 3.4.2. (MHC exposure 200 ms, gain 5; DAPI exposure 100 ms, gain 1). Every picture was analyzed using a self-made macro for the open-source software ImageJ 1.53k. Simplified pictures were color balanced (MHC), or brightness adjusted (DAPI). Subsequently, the threshold was set to further analyze the particles. Minimum criteria for myotubes were set to 750 µm^2^. Resulting MHC pictures were excluded if the area was calculated ≥ 590,000 µm^2^, meaning the lower threshold level was too low for this to replicate. DAPI pictures were excluded if ≥30% of the picture missed DAPI nuclei. For each image, the total MHC area was calculated (*n* ≥ 22). The mean myotube size per picture (*n* ≥ 22) was calculated by dividing the MHC area by the number of counted objects within an image. Dividing the DAPI area on the MHC area by the median size of all DAPI nuclei within each group led to the number of DAPI nuclei on MHC. Furthermore, dividing the number of DAPI nuclei on MHC by the number of MHC objects resulted in the mean nuclei fusion index for each picture (*n* ≥ 22).

### 2.7. Functional Contraction Analysis

A total of 60.000 C2C12 myoblasts were seeded on 12 mm round coverslips and differentiated as described. Experiments were performed on day 9 of differentiation with Tyrode solution (comprising 140 NaCl, 5.4 KCl, 1.8 CaCl_2_, 2 MgCl_2_, 10 glucose (Sigma-Aldrich), and 10 HEPES; pH 7.4, adjusted with NaOH; Carl Roth Gmbh & Co. KG, Karlsruhe, Germany, if not labeled differently) heated by a circulating water system to keep the cells at approximately 35 °C. Electrical stimulation (2 ms long biphasic pulses, 45 V, 1 Hz) was performed with two platinum electrodes and the Myostim stimulator (Myotronic, Germany). An inverted IX73 microscope equipped with a 20x objective (LUCPLFLN20XPH/0,45) was used to first screen manually and meandering for contracting myotubes. The number of contracting myotubes was counted blinded and normalized to the mean MHC area of the same group giving the contracting myotubes per MHC area (*n* = 18 coverslips). Electrically induced contractions were imaged using a UI-306xCP-M camera (iDS, Germany) and analyzed with the custom made Myocyte online Contraction Analysis (MoCA) software described in detail by Wagdi et al. [20]. The resulting images were captured using the camera and analyzed in real-time, with the absolute average of motion vectors per frame denoted as |V|, being output through the NI 9263 CompactDAQ (NI, Austin, U.S.) and recorded with Powerlab 8/35 and LabChart 8.1.16 software for further analysis. The software and Powerlab were used to trigger electrical pacing and record the stimulation parameters simultaneously to the MoCA traces. Each contracting myotube was measured twice. LabChart was used to identify 10 following contractions and relaxations. The time between electrical stimulus and the maximum height of contraction was analyzed as the time to peak (*n* ≥ 15 myotubes). Further, the time between the maximum contraction and maximum relaxation peak was defined as contraction duration (*n* ≥ 15 myotubes).

### 2.8. Bulk RNA-Sequencing

The read count files (GSE145480) were provided by Börsch A et al. [21]. Samples of 8 young mice (8 month) were compared to 9 old mice (28 month) to analyze age-dependent expressional differences. The differential expression (DE) analysis was performed with DESeq2 (version 2.11.40.6, lfcThreshold = 0, alpha = 0.1, minimum count = 0.5). Significantly differential regulated genes were selected by a Benjamini–Hochberg-adjusted *p*-value < 0.05 and log2-fold changes ≥0.75. An exemplary RNA-seq analysis vignette is provided as an R notebook from our group elsewhere [22]. The intergroup DE genes were plotted in a principal component analysis. Further, the 30 most significantly different expressed genes were displayed in a heatmap using the pheatmap package (version 1.0.12). Out of these 30 genes, only the genes that had higher expression in old mice (*n* = 23) were used for gene ontology by the Human Phenotype Ontology, EnrichR databank [23]. The whole DE genes were displayed graphically using the EnhancedVolcano package (version 1.8.0).

### 2.9. Single Cell RNA-Sequencing

The transcriptome-wide analysis of human muscle cells at a single cell level was based on a previously published dataset (GSE130646 from [24]). Here, M. vastus lateralis (*n* = 4) from healthy humans was dissected and droplet-based scRNA-seq was performed. Sequencing data was aligned to the human genome hg19. Data with at least 500 unique molecular identifiers (UMIs), log10 genes per UMI > 0.8, >250 genes per cell and a mitochondrial ratio of less than 20% were extracted, normalized and integrated using the Seurat package v3.0 in R4.0.2. After quality control and integration, we performed a modularity-optimized Louvain clustering with the resolution “1.4”, leading to 40 distinct clusters in the human dataset. Subsequently, we performed the labelling for these 40 clusters manually with established key marker genes (from [24]). A subsequent R-package was dittoSeq (1.2.697). An exemplary notebook from our group for scRNA-Seq can be found elsewhere [25].

### 2.10. Statistics and Graphs

The D’Agostino–Pearson test for normality was performed on every data set. If passed, a normality distribution was assumed, and one-way ANOVA was performed. A Tukey post-hoc test was used to correct for multiple comparisons. If the normality test was not passed, a Kruskal–Wallis test was performed. To correct for multiple comparisons, Dunn’s test was used. Significant differences are depicted as * (*p* < 0.05), ** (*p* < 0.01), *** (*p* < 0.001) and **** (*p* < 0.0001). Statistics were calculated and graphs were designed using GraphPad Prism 9.4.1 (GraphPad Software, Inc., CA, USA), Biorender.com and R 4.0.3.

## 3. Results

### 3.1. Increased Expression of Foxo3 in Satellite Cells Reflects Aging in Human and Murine Skeletal Muscle

To assess the transcriptome-wide importance of *Foxo3* in human skeletal muscle, we used a single cell sequencing dataset from human M. vastus lateralis [24]. Here, we assigned cell types based on top gene expression using t-distributed stochastic neighbor embedding (t-SNE, Figure 1A). We determined the *FOXO3* expression within the distinct cell populations and identified satellite cells to express *Foxo3* at the highest levels (Figure 1B). To assess whether the enrichment of *Foxo3* was age-dependent, we compared the transcriptome of young (8 month) and old (28 month) murine muscle samples (*n* = 18). In a principal component analysis (PCA), most samples from young mice differed substantially from old murine muscles (Figure 1C). An in-depth analysis on individual gene expression revealed *Foxo3* expression to be significantly upregulated in older mice. Besides *Foxo3*, *Fbxo32*, as its target gene, and *Cdkn1a*, a common marker of aging, showed a differential expression in older mice (Figure 1D). The overall most differentially expressed genes were *Sln*, *Cacna1i* and *Sh3rf2*, all of which were had higher expression in the older group (Figure 1E). Gene ontology analysis was performed for the genes that had higher expression in the older group, revealing these genes to cause skeletal atrophy diseases in humans. (Figure 1F).

### 3.2. AAV-Mediated Foxo3 Knockdown Leads to a Stable and Lasting Gene Knockdown in C2C12 Myoblasts

Since we hypothesize an upregulation of *Foxo3* to be related to muscle disorders, we aimed for a technical *Foxo3* knockdown. Due to the highest *Foxo3* expression in satellite cells, C2C12 myoblasts were used. Originally, this cell line was derived from mice and exhibited behavior such as satellite cells [26]. We used different RNAi-based reagents, subdivided into pre- and parallel application, whereby only AAV-mediated plasmid delivery generated a long-term knockdown (Figure 2A). First, we transfected one siRNA against *Foxo3* through single and double application via lipofectamine within one differentiation protocol, where we observed a short-term *Foxo3* knockdown for single lipofection (up to 3 days, Figure 2B, Appendix A) and double lipofection (knockdown up to Day 6, Figure 2C). In addition, we used AAV-mediated transduction to establish a more stable knockdown (Figure 2D–F). As expected, single lipofections only led to a significant *Foxo3* knockdown on day 1 and day 3 (*p* < 0.0001) (Figure 2B). In comparison, double lipofection reached a successful *Foxo3* knockdown until day 5 (*p* < 0.0001), day 6 (*p* < 0.05) and day 8 (*p* < 0.001), but with a trend of losing knockdown efficiency during differentiation (Figure 2C). Interestingly, we reached a knockdown efficiency of 80% for *Foxo3* by AAV transduction (*p* < 0.01) on day 1, which turned out to not be significant on day 3 and even exceeded the control group from day 5 onwards (*p*<0.0001) on the transcriptional level (Figure 2D). However, on the protein level, the FOXO3 knockdown was successful and long-lasting from day 3 on (*p* < 0.05, *p* < 0.01) (Figure 2E,F).

### 3.3. Foxo3 Knockdown Level and Selection of Groups with Equal Growth Rate

To be consistent in starting conditions, we generated single AAV-mediated FOXO3 knockdown clones by FACS cell sorting and gated by GFP intensity. Selected clones were then screened for *Foxo3* knockdown efficiency compared to C2C12 wildtype (WT) cells. The cells with the highest *Foxo3* knockdown were chosen for further experiments. All control clones, two polyclonal control groups and siFoxo3 K4 and K6 did not reach sufficient knockdown, presumably due to an insufficient transduction success (Figure 3A). To determine growth speed, the cell number of all single clones of the *Foxo3* knockdown (K1, K2, K4 and K6) and control (K8, K9, K10, K11 and K12) were pooled together at different time points. Gompertz growth speed best fit the growth curve (siControl: R^2^ = 0.66, siFoxo3 R^2^ = 0.76). Area under the curve analysis revealed no overall group difference in growth behavior in control vs. siFoxo3 cells (*p* = 0.97) (Figure 3B). For a better individualization of the groups, which is important to choose the two main groups for comparison, we analyzed the growth speed using an electrical impedance-based proliferation assay displayed as the cell index. Only siFoxo3 K2 and siControl K8 showed a comparable growth speed. Therefore, we assumed consistent conditions upon confluence for differentiation (Figure 3C,D). The following experiments continued with these two groups, in the following labelled as siControl and siFoxo3.

### 3.4. Foxo3 Knockdown Myotubes Display a Reduced Myotube Size and Lower Myoblast Fusion

Next, cells were morphologically analyzed. Compared to the control group, a lower number of total myotubes and smaller myotubes were observed in the *Foxo3* knockdown myotubes (Figure 4A–C, Appendix A). Analysis of the MHC positive area revealed a smaller area in siFoxo3 compared to siControl pooled (*p* < 0.0001) and siControl (*p* < 0.0001) groups. But within the control groups, a significant (*p* < 0.0001) difference in the MHC positive area was found. Calculated mean myotube size revealed a decreased myotube size in siFoxo3 compared to siControl pooled (*p* < 0.0001) and siControl (*p* < 0.001). Calculated nuclei fusion indexes in both control groups indicated significantly less nuclei per myotube in siFoxo3 (*p* < 0.0001).

### 3.5. Foxo3 Knockdown Leads to Comparable Contracting Myotubes per MHC Area and Prolonged Contraction Duration and Time-to-Peak

To test whether the morphological and transcriptional effects of a *Foxo3* knockdown also effected the biomechanical properties of myotubes, an electrical stimulus was applied to the control group and *Foxo3* knockdown myotubes to trigger contractions. The number of contracting myotubes was counted and normalized to the mean MHC area of each group. WT C2C12 were used as the positive control. The WT displayed lower contracting myotubes per MHC area compared to siControl pooled and siFoxo3 (*p* < 0.0001). Remarkably, no contractions were observed for siControl. SiControl pooled displayed no differences in contracting myotubes per MHC area compared to siFoxo3 (Figure 5A). The kinetic parameters time to peak and contraction duration revealed no differences between WT and siControl pooled, while siFoxo3 revealed a longer contraction duration as well as time to peak compared to WT (*p* < 0.0001) and siControl pooled (*p* < 0.01) (Figure 5B,C).

### 3.6. Differentiation Marker and Genes upon Myogenic Differentiation

After the morphometric analysis, we characterized the differentiating myotubes on the molecular level. At first, we analyzed *Fbxo32* as a direct target of FOXO3 [14], which had lower expression in siFoxo3 in earlier differentiation but displayed a higher expression in later differentiation (Figure 6A). Intriguingly, the *Foxo3* knockdown led to lower levels of *Myod1*, which is an important factor for cell cycle exit in differentiation (Figure 6B) [27]. Moreover, another differentiation parameter was reduced within the same group: *Myog*, an important factor for myoblast fusion [28], was expressed significantly lower in siFoxo3 (*p* < 0.0001) from day 3 onwards (Figure 6C). We observed the activator of DNA damage response, *ATM* [29], to show higher expression in siFoxo3 from day 5 onwards (*p* < 0.0001) (Figure 6D). Furthermore, previous differentiation stages showed lower *Tp53* expression until day 3 (*p* < 0.01) and higher expression from day 7 onwards (*p* < 0.0001) in *Foxo3* knockdown myotubes (Figure 6E). The same dynamic expression was observed for *Foxo1*, another member of the same forkhead transcription factor family (Figure 6F) [12].

## 4. Discussion

Sarcopenia and accompanying diseases lead to a reduced quality of life [5], and potentially limit life expectancy due to falling [3] and prolonged hospitalization [4]. Increased hospital costs cause a worldwide socioeconomic burden [6]. Regrettably, current therapeutical concepts such as nutritional supplementation require long-lasting applications (up to 13 weeks) [10]. Besides, physical activity, another established therapeutical option, is problematic in hospitalized patients because over 80% of the time in hospital is spent lying in bed [9]. Therefore, novel, fast and easily applicable therapeutic concepts are required.

Based on this circumstance, we aimed to evaluate the existence of an ageing dependent *Foxo3* expression. We identified satellite cells to be the main source of *Foxo3* in human muscles. Differential gene expression in muscles from younger compared to older mice showed the expected age-dependent increase in *Foxo3* mRNA. Concordantly, elevated nuclear FOXO3 protein and lower cytosolic phosphorylated FOXO3 (inactive FOXO3) levels were observed in older women (85 ± 1 years) compared to younger women (24 ± 2 years) [30]. Furthermore, we identified the transcriptome of aging mice that is related to aging-associated skeletal muscle diseases in humans. In line with this, a network-based analysis of blood samples from Duchenne muscular dystrophy also identified the FoxO signaling to play a role compared to healthy controls in silico [31]. Additionally, reports highlight the role of FOXO3 in human muscle wasting and atrophy [32].

The potential of FOXO3 as a target was clarified by in vivo studies in mice, where L-carnitine treatment led to phosphorylation of FOXO3 by AKT and overall decreased FOXO3 and FBXO32 protein levels, correlated with an increased gastrocnemius muscle mass [33]. However, no beneficial effect on muscle strength was observed after isolated L-carnitine supplementation over 24 weeks in healthy older females (≥65 years) [34]. Therefore, we investigated the effect of an isolated *Foxo3* knockdown in an AKT-independent mechanism to possibly bypass supplementation deficiencies.

According to our hypothesis, IGF-1 inhibits *Fbxo32* expression via an upregulated PI3K/AKT pathway and suppression of FOXO3 by phosphorylation (Figure 7) [14]. Our *Foxo3* knockdown, which we confirmed on the protein level to be stable, reduced the *Fbxo32* expression until day 3, resulting in higher *Fbxo32* expression in *Foxo3* knockdown myotubes. We also observed an increased expression of *Foxo1* during the differentiation process. We hypothesize that the upregulation of *Foxo1* is compensatory because it is also an activator of *Fbxo32*, which might be an explanation for the missing downregulation of *Fbxo32* (Figure 7) [14]. This hypothesis is supported by a study that transduced an unspecific dominant negative FoxO protein, the *Foxo3* DNA binding domain (highly homologous to *Foxo1* and *Foxo4*), in mice in vivo and observed a reduction in *Fbxo32* and an increased cross-sectional area of M. tibialis anterior and M. extensor digitorum [35]. Furthermore, transfecting a MiR-1290 to induce a *Foxo3* knockdown via AKT activation lead to an increased myotube size in vitro [36]. Here, the influence on FOXO1 remains undetermined, but might also be influenced due to the behavior of AKT to also phosphorylate and consequently inactivate FOXO1 [37]. Intriguingly, the expression of *Fbxo32* appeared to be induced one observation point prior to the induction of *Foxo1* expression. We hypothesized other signaling pathways to be important, such as the AMP kinase- [38] or the p38 MAP kinase-pathway [39], which are both described to increase the *Fbxo32* expression in vitro.

*Foxo3* was observed to induce the expression of *Myod1* in C2C12 myoblasts (Figure 7). Consequently, *Foxo3* knockdown led to myogenic differentiation defects, indicated by smaller myotubes [40]. This confirms that the literature is consistent with what we observed in morphological analyses, i.e., smaller myotubes and on the transcriptional level by an overall lower *Myod1* expression after *Foxo3* knockdown. Moreover, myotubes were formed again by adding a MYOD1 construct to the myoblasts [40]. In its function, *Myod1* is involved in conversion of non-muscle cells into myoblasts, but also withdrawal of the cell cycle, a basic condition for myogenic differentiation [27]. Therefore, we hypothesize that our triggered specific and AKT-independent FOXO3 knockdown inhibited its own initiation of differentiation. Furthermore, it is necessary to discuss whether a *Myod1* overexpression could rescue the differentiation defects. Therefore, in vitro studies showed that pig embryonic fibroblasts were transdifferentiated into skeletal muscle cells by a *Myod1* overexpression in combination with various signaling molecules (FGF2, SB431542, CHIR99021, forskolin). However, isolated *Myod1* overexpression failed to generate muscle cells [41]. Another study showed similar results for human pluripotent stem cells differentiated into muscle cells by *Myod1* overexpression combined with a gene knockdown of POUF5F1 in vitro [42]. However, myotubes were formed again by adding a MYOD construct in C2C12 myotubes lacking *Foxo3* expression [40]. These results highlight the differential potential of *Myod1* overexpression, as well as the potential lack of effect of isolated overexpression.

In vivo studies in zebrafish revealed *Myog* to play an important role in myoblast fusion. A *Myog* knockout led to a depletion of multinucleated muscle fibers and formation of mononucleated fibers [28]. Our in vitro findings confirm this theory. We observed a decreased myotube size with a highly significant reduction in nuclei per myotube in our *Foxo3* knockdown group. Conversely, additional *Myog* overexpression could mitigate the negative effects of *Foxo3* knockdown. When MSCs overexpressing *Myog* were transplanted into rat gastrocnemius muscle, the time of denervation-induced muscle atrophy was delayed [43].

*Foxo3* itself is of importance for cell integrity [44], indicating that our forced *Foxo3* knockdown led to increased levels of ATM (DNA damage response), mediating an activation of *Tp53* [29]. The link between *Tp53* and *Foxo3* was studied in mouse models. *Tp53* was identified to bind a *Tp53*-responsive element on the *Foxo3* gene in mouse liver cells in vivo [45]. Therefore, increased expression of *Tp53* might lead to a rebound *Foxo3* expression due to the direct induction of gene expression, as we observed during the differentiation process as a response to the long-term *Foxo3* knockdown. Interestingly, we observed a high FOXO3 knockdown on the protein level after AAV treatment. Therefore, we hypothesize that *Foxo3* siRNA inhibited the translation without degradation of the *Foxo3* mRNA [46], but Tp53 does not only mediate the *Foxo3* gene expression. By binding to its response element on the *Myog* promotor, *Tp53* instead inhibits the *Myog* expression in mouse embryonic fibroblast (MEF) in vitro [47]. However, *Myog* expression was significantly downregulated prior to *Tp53* induction. Here, the expression of *Myod1* and *Myog* is coupled because *Myod1* itself promotes the expression of *Myog* in vitro [48]. Therefore, we hypothesized that the absence of *Myod1* in early differentiation does not increase the *Myog* expression, but in later differentiation, it is suppressed by the induction of *Tp53.*

Regarding the morphology, we observed a reduction in myotube size and myoblast fusion to myotubes due to the *Foxo3* knockdown. The cell count for the siFoxo3 group appeared to be higher compared to the control (Figure 4A–C). Here, FOXO3 is described to increase the rate of apoptosis [49]. Thus, due to lower *Foxo3* levels (siFoxo3), the rate of apoptosis is reduced, resulting in a higher number of surviving myoblasts compared to the control group. We hypothesized that siFoxo3 exhibited lower myoblast differentiation and lower apoptosis, leading to a differentiated state of smaller myotubes and more resident myoblasts.

We were able to trigger and detect more contracting myotubes in the siControl pooled and siFoxo3 group compared to wildtype. In contrast, we were not able to detect any sufficient contractions in the monoclonal control group. Hence, time to peak as well as contraction duration could not be measured in this group.

Our hypothesis is that the monoclonal control group has a higher *Foxo3* level compared to the other control group (Figure 3). Higher levels of *Foxo3* are described to lead to an increased rate of apoptosis [49], which might limit the functional outcome. In contrast, the *Foxo3* knockdown myotubes display a longer time to peak and contraction duration. It has been shown that a longer time to peak indicates decelerated contraction dynamics in myotubes from amyotrophic lateral sclerosis patients derived from induced pluripotent stem cells in vitro [50]. In conclusion, we hypothesize a *Foxo3* equilibrium. Higher *Foxo3* levels lead to loss of functionality because lower levels limit differentiation. We were not only able to detect contractions online in a 2D skeletal muscle cell culture, but also measure parameters that permit a statement on the kinetic properties even though this does not allow any conclusion to be drawn on the actual contraction force given by a 3D tissue [51].

There are limitations to discuss. First, this study uses publicly available in silico (human and mouse) data and murine in vitro data. The findings warrant confirmation in in vivo models and human studies. Here, the muscle tissue is mainly consisting of already-differentiated muscle cells. We used a previous AAV transduction prior to differentiation. Following investigations should focus on the effect of *Foxo3* knockdown after myoblast differentiation. Moreover, there are further investigations on a combined and unspecific *Foxo3* and *Foxo1* knockdown needed to exclude a compensatory regulation. Additionally, further investigations should also aim for *Foxo3* overexpression experiments, targeting the question of a higher myoblast differentiation and fusion rate due to FOXO3.

## 5. Conclusions

In conclusion, a *Foxo3* knockdown prior to differentiation seems to inhibit its own initiation of differentiation through decreased *Myod1* expression. Moreover, lower *Foxo3* levels lead to an *ATM*-dependent *Tp53* induction that inhibits *Myog* expression and consequently limits myoblast fusion to myotubes. Given the presence of AAV therapies in ongoing clinical trials (250 trials in 2021 [52]), the demonstrated AAV-mediated FOXO3 treatment might depict a starting point for further studies on the genetic treatment of sarcopenia. Although our hypothesis was surprisingly disproved, this could be due to the focus on differentiated and not-differentiated muscle fibers. A new approach would examine a *Foxo3* knockdown on differentiated muscle cells.

## Figures and Tables

**Figure 1 cells-12-02167-f001:**
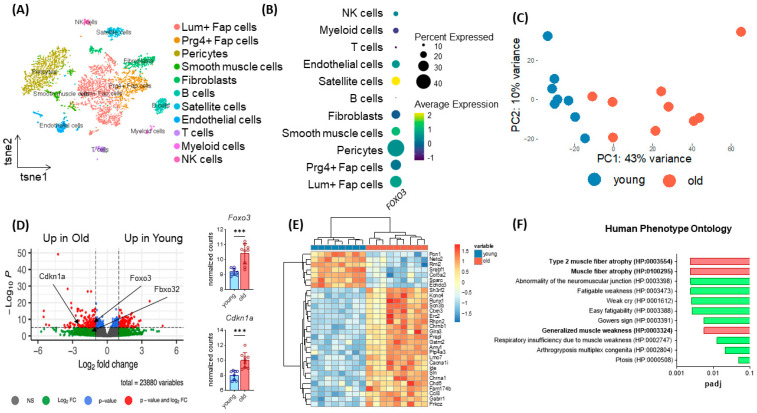
Age-dependent upregulation of *Foxo3* reflecting sarcopenia-related phenotypes. (**A**) TSNE plot displaying the different cell types in human M. vastus lateralis samples (*n* = 4, GSE130646) clustered by distinct expressional profiles. (**B**) Dot plot reflecting *FOXO3* expression in eleven previously clustered cell types. Satellite cells show the highest *Foxo3* expression. (**C**) PCA Plot reflecting the general expression pattern of young and old mice upon aging. (**D**) Volcano plot of (*n* = 17 mice) split into young (8 months, *n* = 8) and old (28 months, *n* = 9) individuals of M. gastrocnemius muscle displaying upregulation of *Foxo3* expression in old mice (GSE145480). *** *p* < 0.001. (**E**) Heatmap displaying the 30 most differently expressed genes. (**F**) Gene ontology (Human Phenotype Ontology, EnrichR) reflecting upregulated genes associated with muscle atrophy diseases in humans (padj < 0.01).

**Figure 2 cells-12-02167-f002:**
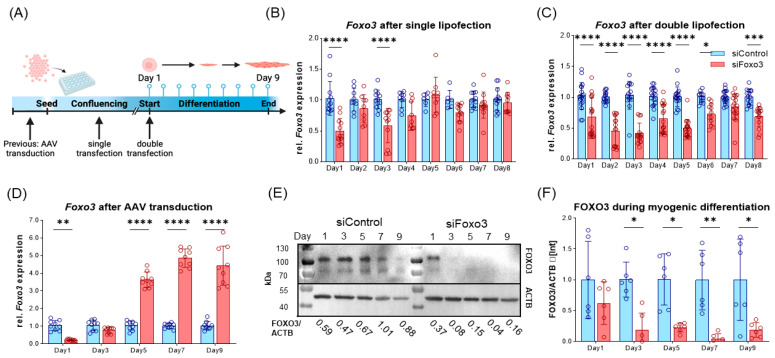
Long-term FOXO3 knockdown can be achieved by AAV-mediated siRNA transduction. (**A**) Timeline visualizing experimental design for single and double lipofection as well as for AAV-mediated transduction. For single lipofection, siRNA was transfected during myoblast confluence; for double transfection, an additional second transfection was applied when switched to low serum conditions. AAV transduction was performed prior to differentiation. (**B**) Single lipofection of a siRNA only led to significant (*p* < 0.0001) *Foxo3* knockdown on day 1 and day 3 (*n* ≥ 6). (**C**) Double lipofection showed a stable knockdown efficiency within the first six days (*p* < 0.0001 and *p* < 0.05), with fluctuating expression in the following days indicating a reduced *Foxo3* knockdown efficiency (*n* =≥ 12). (**D**) AAV-mediated *Foxo3* knockdown showed significant (*p* < 0.01) 80% knockdown efficiency on day 1, which is already lost on day 3, and a 3.6-fold increased (*p* < 0.0001) *Foxo3* rebound expression on day 5 as a stable trend for the following days (*n* = 9). (**E**) Representative Western Blot of FOXO3 protein upon differentiation. (**F**) Quantified Western Blot displaying FOXO3 protein level normalized to Actin Beta (ACTB) loading control after AAV-mediated *Foxo3* knockdown showing long-term stable and significant (*p* < 0.05, *p* < 0.01) knockdown on protein level upon differentiation (*n* = 6). Mean ± SD. Significant differences are depicted as * (*p* < 0.05), ** (*p* < 0.01), *** (*p* < 0.001) and **** (*p* < 0.0001).

**Figure 3 cells-12-02167-f003:**
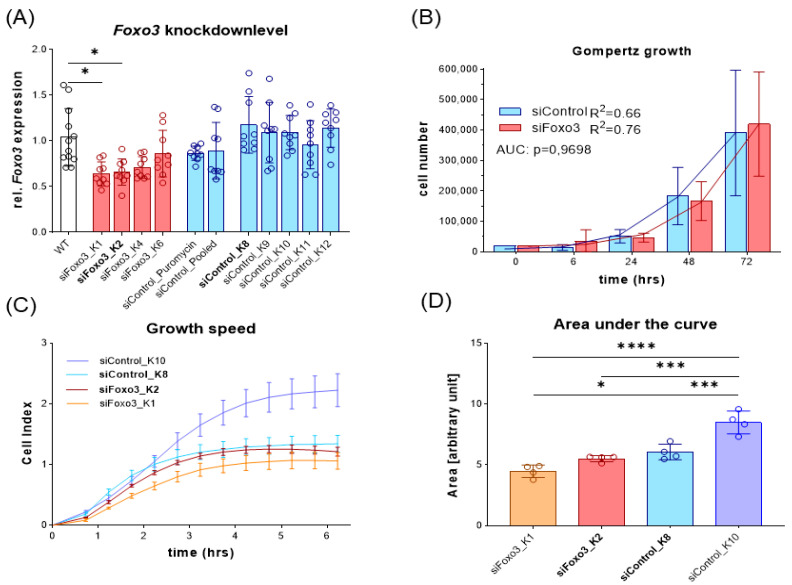
*Foxo3* knockdown displayed no overall pooled group differences while siControl K8 (light blue) and siFoxo3 K2 (red) individually showed equal growth rates in myoblast stadium (**A**) The relative *Foxo3* level compared to the C2C12 WT from different FACS sorted mono- and polyclonal clones. *Foxo3* clone one (K1) and two (K2) show significant Foxo3 knockdown (*p* < 0.05), while the other clones as well as single and polyclonal control groups did not express *Foxo3* differently compared to C2C12 WT cells (*n* ≥ 9). All statistically significant comparisons to the WT cells are displayed. (**B**) Gompertz growth best fitted the growth behavior of *Foxo3* and the control group. Different clones were pooled and the area under the curve analysis (AUC, students t-test) showed equal growth speed within the first 72 h (*n* ≥ 4). (**C**) Growth speed for individual groups for the first 6 h after seeding (*n* = 4). (**D**) Area under the curve analysis from (**C**) revealing siControl K8 and siFoxo3 K2 to be the best matching clones (*n* = 4). Mean ± SD. Significant differences are depicted as * (*p* < 0.05), *** (*p* < 0.001) and **** (*p* < 0.0001).

**Figure 4 cells-12-02167-f004:**
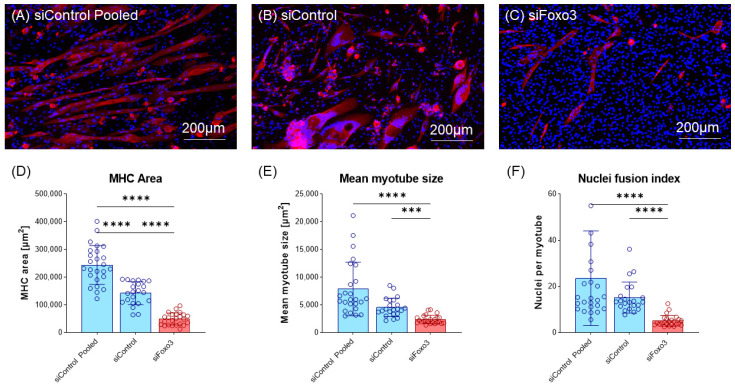
Knockdown of *Foxo3* leads to impaired differentiation, resulting in smaller myotubes. (**A**–**C**) Representative MHC staining for Control and *Foxo3* knockdown groups. (**D**) Quantified MHC area (*n* ≥ 22) displays the *Foxo3* knockdown group having a significantly (*p* < 0.05, *p* > 0.0001) smaller MHC positive area compared to the two control groups. Within the control groups, siControl pooled displayed a higher MHC positive area (*p* < 0.001) compared to siControl. (**E**) An estimation model for the individual single myotube size revealed a smaller myotube size (*p* < 0.05) for *Foxo3* knockdown myotubes compared to siControl (*n* ≥ 22). (**F**) An estimation model for a single myotube nuclei fusion Index displayed a significantly (*p* < 0.0001) reduced number of nuclei per myotube for *Foxo3* knockdown myotubes compared to both control groups (*n* ≥ 22). Scale bar represents 200 µm. Mean ± SD. Significant differences are depicted as *** (*p* < 0.001) and **** (*p* < 0.0001).

**Figure 5 cells-12-02167-f005:**
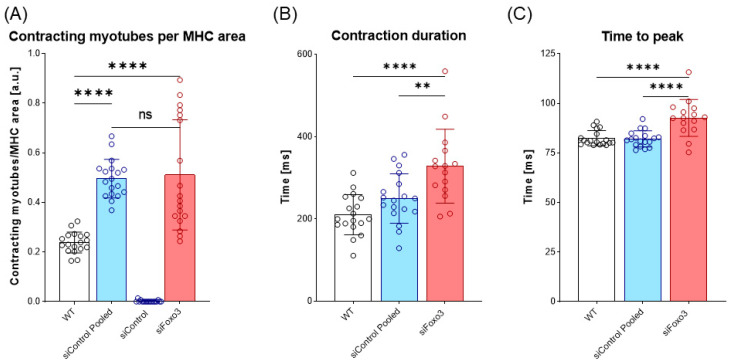
Prolonged contraction duration as well as time to peak after *Foxo3* gene knockdown (**A**) Contracting myotubes per MHC area (*n* = 18). C2C12 wildtype (WT) was used as the positive control. SiControl pooled and siFoxo3 share a non-significant relation and display more contracting myotubes per MHC area compared to siControl (*p* < 0.0001). (**B**) The time between maximum contraction and maximum relaxation is defined as contraction duration (*n* ≥ 15). WT and siControl pooled showed no different behavior, while siFoxo3 revealed a longer contraction duration compared to siControl pooled (*p* < 0.01) and WT (*p* < 0.0001). (**C**) Time between electrical stimulus and maximum contraction as time to peak (*n* ≥ 15). WT and siControl pooled indicated no differences in time to peak while siFoxo3 showed a longer time to peak compared to both groups (*p* < 0.0001). Mean ± SD. Significant differences are depicted as ** (*p* < 0.01) and **** (*p* < 0.0001). ns = non significant.

**Figure 6 cells-12-02167-f006:**
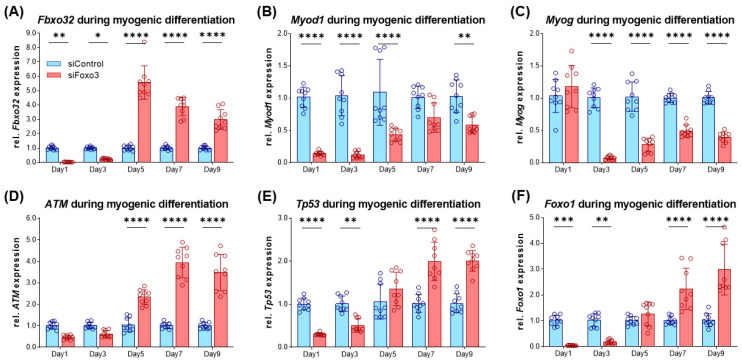
Impaired differentiation is accompanied by lower *Myod1* and *Myog* expression levels in *Foxo3* knockdown myotubes (**A**) *Fbxo32* was expressed lower within the first three days (*p* < 0.01, *p* < 0.05) and was expressed higher from day 5 on (*p* < 0.0001) in siFoxo3 cells. (**B**) *Myod1* was significantly reduced within the first five days (*p* < 0.0001) and on day 9 (*p* < 0.01). (**C**) *Myog* expression was significantly (*p* < 0.0001) reduced during myogenic differentiation in the siFoxo3-group. (**D**) ATM expression was significantly higher in the *Foxo3* knockdown group from day 5 onwards. (**E**) *Tp53* was expressed lower in the *Foxo3* knockdown group within the first three days (*p* < 0.01, *p* < 0.0001), and was expressed higher compared to the control group expression from day seven onwards (*p* < 0.0001). (**F**) Foxo1 was expressed lower in the *Foxo3* knockdown group within the first three days (*p* < 0.01, *p* < 0.001) and was higher compared to control group expression after day seven (*p* < 0.0001), analogous to *Tp53*. Mean ± SD. Significant differences are depicted as * (*p* < 0.05), ** (*p* < 0.01), *** (*p* < 0.001) and **** (*p* < 0.0001).

**Figure 7 cells-12-02167-f007:**
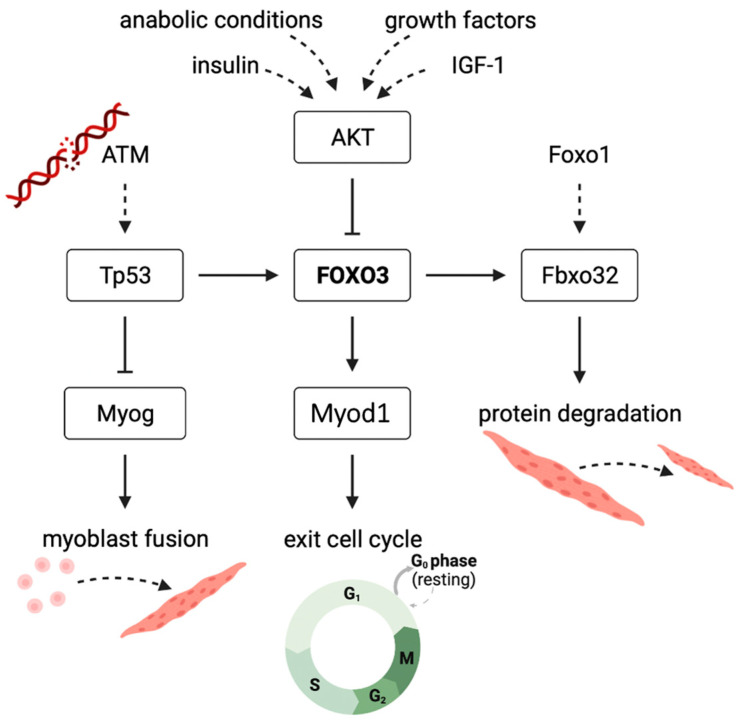
Suggested mechanism displaying the effect of *Foxo3* gene knockdown on myogenic differentiation in C2C12 myotubes in vitro FOXO3-induced transcription of *Fbxo32* to cause protein degradation. FOXO3 is further inhibited by the PI3K-AKT pathway in anabolic conditions. Knockdown of FOXO3 suppresses the expression of *Myod1*, an important player for the initiation of myogenic differentiation. Increased *ATM* levels are caused by DNA damage due to a lack of protective FOXO3. ATM expresses *Tp53* to increase FOXO3, and it inhibits *Myog* that is needed for myoblast fusion to myotubes.

**Table 1 cells-12-02167-t001:** Primer sequences.

Gene	Forward Primer (5′→3′)	Reverse Primer (5′→3′)
*Foxo3*	CGC TGT GTG CCC TAC TTC	CCC GTG CCT TCA TTC TGA
*Fbxo32*	CAG CTT CGT GAG CGA CCT C	GGC AGT CGA GAA GTC CAG TC
*Myod1*	GCC CGC GCT CCA ACT GCT CTG	CCT ACG GTG GTG CGC CCT CTG
*Myog*	CAT CCA GTA CAT TGA GCG CCT	GAG CAA ATG ATC TCC TGG GTT
*Tp53*	CAC AGC ACA TGA CGG AGG TC	TCC TTC CAC CCG GAT AAG ATG
*ATM*	GAG TGA GAC GGG CTG TTA CC	CAT GCT GCC TCC TTC TTT TC
*Foxo1*	GCG GGC TGG AAG AAT TCA AT	TCC AGT TCC TTC ATT CTG CA
*Gapdh*	CTC CCA CTC TTC CAC CCT CG	GCC TCT CTT GCT GAG TGT CC

## Data Availability

The datasets used and/or analyzed during the current study are available from the corresponding author on reasonable request.

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
