# Peer review of "Foxo3 Knockdown Mediates Decline of Myod1 and Myog Reducing Myoblast Conversion to Myotubes"

_cells, 2023, doi:10.3390/cells12172167_

Round 1

Reviewer 1 Report

In the study conducted by Benjamin et al., the authors performed AAV-mediated knockdown of FOXO3 in C2C2 Cells. They found that after stable knockdown of FOXO3, there is a reduction in myoblast conversion to myotubes. While their work is intriguing, the current data does not seem sufficient to support their conclusions. I have concerns regarding the presented data, which I've outlined below.

1: The figure quality in this paper is poor, making it difficult to interpret the results accurately.

  1. In Figure 2B-C, the efficiency of Foxo3 knockdown appears to be very low with the AAV method used. I am not convinced by the AAV knockdown experiment presented in this paper. Other knockout methods like Cas9 knock out or redesign primer knockdown of FOXO3 should be performed to further confirm the data presented in Figure 4A-C.

  2. In Figure 3B and 3C, although the cell number shows no difference, the cell growth rate seems to be different. This inconsistency requires clarification.

  3. In Figure 4C, it appears that the cell number in the FOXO3 knockdown cell line is different from the other groups. They found MHC to be less in knockdown cells. An overexpression FOXO3 experiment should be performed to further confirm these data since it is the main conclusion in this paper and should be examined carefully.

5: In Figure 6, impaired differentiation is accompanied by lower Myod1 and Myog expression levels in Foxo3 knockdown myotubes. It is essential to investigate whether Myod1 or Myog overexpression can rescue the FOXO3 knockdown effect.

6: In patients with muscular dystrophy, what is the expression pattern of FOXO3? This information may provide valuable insights into the relevance of the findings in the context of human disease.

7: As the authors mentioned, FOXO3 has been extensively studied in zebrafish and mice. Therefore, the significance of this cellular study may not be as evident. I suggest focusing more on investigating FOXO3 in patient samples to strengthen the clinical relevance of the research.

Author Response

In the study conducted by Benjamin et al., the authors performed AAV-mediated knockdown of FOXO3 in C2C2 Cells. They found that after stable knockdown of FOXO3, there is a reduction in myoblast conversion to myotubes. While their work is intriguing, the current data does not seem sufficient to support their conclusions. I have concerns regarding the presented data, which I've outlined below.

1: The figure quality in this paper is poor, making it difficult to interpret the results accurately.

We increased the resolution of the figures substantially and hope to meet the expectations of the reviewer.

In Figure 2B-C, the efficiency of Foxo3 knockdown appears to be very low with the AAV method used. I am not convinced by the AAV knockdown experiment presented in this paper. Other knockout methods like Cas9 knock out or redesign primer knockdown of FOXO3 should be performed to further confirm the data presented in Figure 4A-C.

Indeed, the Foxo3 knockdown in Figure2B/C appears to be low, especially in later time point (day 5 and later). Nevertheless, the experiments in Figure 2B/C were performed via lipofection and not via AAVs. Lipofection is known for time limiting effects (2-3 days) on cells. Due to this time restriction (knockdown only detectable for 3 days (single transfection, Figure 2B) and for 6 days (double transfection, Figure 2C)) we decided to perform long lasting AAV knockdown experiments (Figure 2D-F). We could detect a Foxo3 knockdown on protein level from day 3 up to day 9 (Figure 2 E, F). The Foxo3 knockdown on protein level was highly significant and we could achieve a knockdown efficiency of up to 95%. We clarified this in the manuscript (line 291-299)

First, we transfected one siRNA against Foxo3 by single and double application via lipofectamine within one differentiation protocol, where we observed a short-term Foxo3 knockdown for single lipofection (up to 3 days, Figure 2B, Suppl. Figure 1) and double lipofection (knockdown up to Day 6, Figure 2C). In addition, we used AAV-mediated transduction to establish a more stable knockdown (Figure 2 D-F). As expected, single lipofections only led to a significant Foxo3 knockdown on day 1 and day 3 (p<0.0001) (Figure 2B). In comparison, double lipofection reached a successful Foxo3 knockdown until day 5 (p<0.0001), day 6 (p<0.05) and day 8 (p<0.001) but with a trend of losing knockdown efficiency during differentiation (Figure 2C).

In Figure 3B and 3C, although the cell number shows no difference, the cell growth rate seems to be different. This inconsistency requires clarification.

We thank the reviewer for this comment. The differences between overall group differences and individual cell line differences in terms of growth speed are clarified in the text (line 333-340). In this manuscript, we focused on the comparison of siControl K8 and siFoxo3 K2, which showed no differences in growth rate (Figure 3C, D). The comparable growth rate is important for the differentiation experiments shown later in the manuscript. We clarified this section in the manuscript.

Area under the curve analysis revealed no overall group difference in growth behavior in control vs. siFoxo3 cells (p=0.97) (Figure 3B). For a better individualization of the groups, which is important to choose the two main groups for comparison, we analyzed the growth speed by an electrical impedance-based proliferation assay displayed as cell index. Only siFoxo3 K2 and siControl K8 showed a comparable growth speed. Therefore, we assumed consistent conditions upon confluence for differentiation (Figure 3C and 3D). Following experiments continued with these two groups, in the following labelled as siControl and siFoxo3.

In Figure 4C, it appears that the cell number in the FOXO3 knockdown cell line is different from the other groups. They found MHC to be less in knockdown cells. An overexpression FOXO3 experiment should be performed to further confirm these data since it is the main conclusion in this paper and should be examined carefully.

Indeed, it seems that more cells are visible in Figure 4C. For every group, 60.000 cells were sed, incubated for 48h and differentiated for 9 days afterwards. Due to the lower fusion of myoblasts in the siFoxo3 group, more single nuclei are visible in Figure 4C. In Figure 4A/B, more MHC-positive cells with multiple and denser nuclei are visible which could give the impression of less cells. Additionally, Foxo3 drives apoptosis. Therefore, lower Foxo3 levels led to lower apoptosis and more survived myoblast consequently (line 525-532). We added and discussed the Foxo3 overexpression experiment in the limitation part of this manuscript (line 557-559)

Regarding the morphology, we observed a reduction in myotube size and myoblast fusion to myotubes. The cell count for the siFoxo3 group, appeared to be higher compared to the control (Figure 4A-C). Here, Foxo3 is described to increase the rate of apoptosis [46]. Thus, due to lower Foxo3 levels (siFoxo3), the rate of apoptosis is reduced, resulting in a higher number of surviving myoblasts than control. We hypothesized that siFoxo3 exhibited lower myoblast fusion and lower apoptosis, leading to a differentiated state of smaller myotubes and more resident myoblasts.

Additionally, further investigations should also aim for Foxo3 overexpression experiments, targeting the question of a higher myoblast differentiation and fusion rate due to FOXO3.

5: In Figure 6, impaired differentiation is accompanied by lower Myod1 and Myog expression levels in Foxo3 knockdown myotubes. It is essential to investigate whether Myod1 or Myog overexpression can rescue the FOXO3 knockdown effect.

We agree with the reviewer that Myod1 and Myog are very important for myogenic differentiation. We further discussed the transdifferential potential of Myod1 overexpression (line 487-497) and the protective character of Myog (line 503-506).

Furthermore, it is necessary to discuss whether a Myod1 overexpression could rescue the differentiation defects. Therefore, in vitro studies showed that pig embryonic fibroblasts were transdifferentiated into skeletal muscle cells by a Myod1 overexpression in combination with various signaling molecules (FGF2, SB431542, CHIR99021, forskolin). However, isolated Myod1 overexpression failed to generate muscle cells [40]. Another study showed similar results for human pluripotent stem cells differentiated into muscle cells by Myod1 overexpression combined with a gene knockdown of POUF5F1 in vitro [41]. However, myotubes were formed again by adding an MYOD construct in C2C12 myotubes lacking Foxo3 expression [39]. These results highlight the differential potential of a Myod1 overexpression but also the potential lack of effect of an isolated overexpression.

Conversely, additional Myog overexpression could mitigate the negative effects of Foxo3 knockdown. When MSCs overexpressing Myog were transplanted into rat gastrocnemius muscle, the time of denervation-induced muscle atrophy was delayed [42].

6: In patients with muscular dystrophy, what is the expression pattern of FOXO3? This information may provide valuable insights into the relevance of the findings in the context of human disease.

There is rare information about muscular dystrophy and the link to FOXO3. We added and discussed a publication, that identified a role of Foxo signaling in patients with muscular dystrophy compared to healthy controls in silico (line 446-449). This also supports our findings of the connection between the aging transcriptome of mice to display a composition related to skeletal muscle diseases in human.

Furthermore, we identified the transcriptome of aging mice which is related to aging-associated skeletal muscle diseases in humans. In line with this, a network-based analysis of blood samples from Duchenne muscular dystrophy also identified the FoxO signaling to play a role compared to healthy control in silico [31]. Additionally, reports highlight the role of FOXO3 in human muscle wasting and atrophy [32].

7: As the authors mentioned, FOXO3 has been extensively studied in zebrafish and mice. Therefore, the significance of this cellular study may not be as evident. I suggest focusing more on investigating FOXO3 in patient samples to strengthen the clinical relevance of the research.

In this study we wanted to investigate the long-term effects of an isolated Foxo3 knockdown at the morphological, functional, and molecular level. Our in vitro approach allows us to directly investigate the effects of an AAV mediated RNAi induced Foxo3 knockdown on myoblasts and their differentiation into myotubes. Furthermore, it allows us to analyze contraction time on a single cell level.

In order to correlate our findings to the recent data we used mainly in vitro and in vivo mouse studies, but also results from human investigations (line 443-445, 454-455). We furthermore added significance of human studies in the limitation part of this manuscript (line 553).

Reviewer 2 Report

The authors investigated the effects of FOXO3 knockdown on the myotube formation of C2C12 cells. They established a line of AAV-mediated FOXO3-knockdown C2C12 myoblasts at the protein level. In the ensuing experiments, the cells exhibited low differentiation potential with low expression levels of Myod1 and myogenin, and with high expression levels of Fbxo32, Tp53 and Foxo1. While the results are somewhat interesting, several issues with the study and manuscript need to be addressed.

The authors explain that the Fbxo32 was upregulated at a later stage in the FOXO-knockdown cells. While the level of Foxo1 expression on Day 5 was similar to the expression of siControl (K8), the expression of Fbxo32 was already significantly upregulated at the same point of time. Why was this?

Figure 7:

A significant downregulation of myogenin was observed on Day 3. ATM and Tp53, on the other hand, were both detectably upregulated on Day 5. Was the myogenin downregulated by the upregulation of the ATM-Tp53 axis?

Figure 4B:

The shape of the MHC-positive cells among the siControl cells (K8) is clearly abnormal. Does this abnormality suggest that FOXO3 may be essential for the normal myogenic differentiation? The authors should explain why the MHC-positive cells among the siControl cells (K8) are abnormally shaped.

Figure 6:

Does “siControl” refer to siControl pooled cells? If not, why did the authors use a single clone of siControl as a control? The siControl K8 exhibits an abnormality in the myotube formation.

Author Response

The authors investigated the effects of FOXO3 knockdown on the myotube formation of C2C12 cells. They established a line of AAV-mediated FOXO3-knockdown C2C12 myoblasts at the protein level. In the ensuing experiments, the cells exhibited low differentiation potential with low expression levels of Myod1 and myogenin, and with high expression levels of Fbxo32, Tp53 and Foxo1. While the results are somewhat interesting, several issues with the study and manuscript need to be addressed.

The authors explain that the Fbxo32 was upregulated at a later stage in the FOXO-knockdown cells. While the level of Foxo1 expression on Day 5 was similar to the expression of siControl (K8), the expression of Fbxo32 was already significantly upregulated at the same point of time. Why was this?

We agree to the reviewer, that at day 5 the Foxo1 expression of siControl and siFoxo cells is comparable, while at day 7 Foxo1 and Fbxo32 are overexpressed in siFoxo3 cells. This analysis is based on mRNA measurements via qPCR, which can be highly variable in different time points, especially in differentiation experiments. We see the general trend that Foxo1 and Fbxo32 increase during differentiation and based our hypothesis on this observation. Further signal pathways, such as AMPK or p38 MAPK can be involved and increase the Fbxo32 expression as well. We added this information in the discussion part (line 473-476).

Intriguingly the expression of Fbxo32 appeared to be induced one observation point prior to the induction of Foxo1 expression. We hypothesized other signaling pathways, such as the AMP kinase- [37] or the p38 MAP kinase-pathway [38], that are both described to increase the Fbxo32 expression in vitro.

Figure 7:

A significant downregulation of myogenin was observed on Day 3. ATM and Tp53, on the other hand, were both detectably upregulated on Day 5. Was the myogenin downregulated by the upregulation of the ATM-Tp53 axis?

We thank the reviewer for this important observation. Indeed, there is also a coupled interaction between Myod1 and Myogenin since Myod1 promotes the expression of Myogenin. Therefore, during earlier differentiation, the absence of Myod1 does not increase the Myogenin expression, whereas during later differentiation the Myogenin expression is repressed by Tp53. We discussed this interaction in the manuscript (line 519-523)

However, Myog expression was significantly downregulated prior to Tp53 induction. Here, the expression of Myod1 and Myog is coupled because Myod1 itself promotes the expression of Myog in vitro [47]. Therefore, we hypothesized that the absence of Myod1 in early differentiation does not increase the Myog expression while in later differentiation it is suppressed by the induction of Tp53.

Figure 4B:

The shape of the MHC-positive cells among the siControl cells (K8) is clearly abnormal. Does this abnormality suggest that FOXO3 may be essential for the normal myogenic differentiation? The authors should explain why the MHC-positive cells among the siControl cells (K8) are abnormally shaped.

We agree with the reviewer that the morphology of siControl cells is different to siFoxo3 cells. We saw a high fusion rate of siControl cells in our experiments, leading to large myotubes with a high number of nuclei. We hypothesized that the monoclonal siControl group exhibits a higher Foxo3 level (please see Figure 3) leading to a higher differentiation and nuclei fusion rate compared to a higher Foxo3-induced rate of apoptosis. Consequently, the Foxo3 knockdown group exhibited lower differentiation and nuclei fusion rate and a lower rate of apoptosis. We further discuss the morphology (line 525-532) combined with the functionality (line 538-549). The hypothesis of a Foxo3-mediated myoblast fusion needs further experimental confirmation and is listed in the limitation (line 557-559).

Line 525-532: Regarding the morphology, we observed a reduction in myotube size and myoblast fusion to myotubes due to the Foxo3 knockdown. The cell count for the siFoxo3 group, appeared to be higher compared to the control (Figure 4A-C). Here, FOXO3 is described to increase the rate of apoptosis [48]. Thus, due to lower Foxo3 levels (siFoxo3), the rate of apoptosis is reduced, resulting in a higher number of surviving myoblasts compared to the control group. We hypothesized that siFoxo3 exhibited lower myoblast differentiation and lower apoptosis, leading to a differentiated state of smaller myotubes and more resident myoblasts.

Our hypothesis is that the monoclonal control group has a higher Foxo3 level compared to the other control group (Figure 3). Higher levels of Foxo3 are described to lead to an increased rate of apoptosis [48], which might limit the functional outcome. On the other hand, the Foxo3 knockdown myotubes display a longer time to peak and contraction duration. It has been shown that a longer time to peak indicates decelerated contraction dynamics, in myotubes from amyotrophic lateral sclerosis patients derived from induced pluripotent stem cells in vitro [49]. Concluding, we hypothesize a Foxo3 equilibrium. Higher Foxo3 levels lead to loss of functionality as lower levels limit differentiation. We were not only able to detect contractions online in a 2D skeletal muscle cell culture, but also measure parameters which permit a statement on the kinetic properties even though this does not allow any conclusion on actual contraction force as given by a 3D tissue [50].

Additionally, further investigations should also aim for Foxo3 overexpression experiments, targeting the question of a higher myoblast differentiation and fusion rate.

Figure 6:

Does “siControl” refer to siControl pooled cells? If not, why did the authors use a single clone of siControl as a control? The siControl K8 exhibits an abnormality in the myotube formation.

We chose the siControl K8 clone since it was established in the same way as the siFoxo cells (single clone, FACS sorted). Furthermore, siControl K8 showed a similar growth rate compared to siFoxo3 cells (Figure 3C/D). Nevertheless, the Foxo3 level seems to be altered compared to siControl pooled cells, which could affect the myogenic differentiation (Figure 3A). We discuss this critique in the manuscript (line 525-532, 538-549).

Round 2

Reviewer 1 Report

The author provided a perfect answer to my question. I have no further inquiries and recommend acceptance.

Reviewer 2 Report

The authors have revised their manuscript in response to my comments.